# Robust Stability of Switched-Interval Positive Linear Systems with All Modes Unstable Using the Φ-Dependent Dwell Time Technique

**Qiang Yu** * and **Xiujuan Jiang**

School of Mathematics and Computer Science, Shanxi Normal University, Taiyuan 030032, China; yjssxnu2021@126.com
* Correspondence: yuqiang111111@126.com

**Abstract:** In this study, some stability and robust stability conditions for switched positive linear systems in which all subsystems are unstable in continuous time and discrete time were obtained using the Φ-dependent dwell time technique and the discretized co-positive Lyapunov functions approach, respectively. The co-positive Lyapunov functions constructed in this study are functions of time during the dwell time, and after that, they are independent of time. In addition, the above method was applied to switched-interval positive systems, and corresponding conclusions are presented. The Φ-dependent dwell time technique used in this paper is more effective than the dwell time and mode-dependent dwell time used in other studies. The results are verified with an illustrative example.

**Keywords:** Φ-dependent dwell time; discretized co-positive Lyapunov functions; robust stability; unstable subsystems

**MSC:** 34A38; 37B25; 93D09

## 1. Introduction

Many dynamic systems in the real world are restricted to orthogonal variables, and such systems are usually referred to as positive systems in the literature. Switched positive linear systems (SPLSs), a class of positive systems, consist of a series of positive linear systems and switching signals that control the switching between them. SPLSs have received extensive attention due to their wide applications in control fields, such as mobile robot navigation [1], load forecasting [2], and fuel balancing [3]. In SPLSs, stability is the most basic property of the system. Some important results regarding SPLSs have been obtained in the literature [4–11], especially with respect to stability analysis. The linear co-positive Lyapunov function (LCLF) approach is a very effective approach in the stability analysis of SPLSs because it holds that the state of the systems is positive under natural constraints. The sufficient and necessary conditions for the existence of a common LCLF for SPLSs are mentioned in [12]. The stability of discrete-time SPLSs was analyzed in [13] by using the switched LCLF approach. The multiple LCLF for a given SPLS was established for the first time in [14]. Based on this, the stability analysis problem for SPLSs with average-dwell-time (ADT) switching was investigated and sufficient conditions for both the continuous-time and the discrete-time cases were given in [14].

Most practical switched systems have both stable and unstable subsystems due to disturbances, failures, and possibl changing environmental factors [15]. The problem of exponential stability for SPLSs consisting of both stable and unstable subsystems was investigated in [16]. Sufficient stability conditions were proposed for discrete-time switched delay positive systems with stable and unstable subsystems in [17]. The above conclusions

all concern switched systems with at least one stable subsystem. However, these conclusions are generally not valid when all subsystems are unstable. As is well known, even if all subsystems are unstable, one may carefully switch between them to make the total system asymptotically stable. Using a discretized Lyapunov function approach, a sufficient condition ensuring the asymptotic stability of switched continuous-time systems with all modes unstable is proposed in [18]. The stability of switched linear systems with dwell time was studied by constructing a discretized Lyapunov function in [19]. Some conclusions have been reached [20,21] about SPLSs with uncertainty. In [20], the constructed Lyapunov functions were time-varying during the dwell time and time-invariant afterwards. There are currently few research conclusions about the stability of SPLSs with all modes unstable and uncertainty, which are significant for our research in this paper.

The main objective of this study was to establish the stability and robust stability criteria for a system where all subsystems are unstable in continuous-time and discrete-time cases. The research results in this paper include the following points. First of all, a group of switching signals that make SPLSs with all modes unstable asymptotically stable was found by using discretized co-positive Lyapunov functions and the $\Phi$-dependent dwell time technique. Then, we extended the above method and conclusion to SPLSs with interval uncertainty. Finally, an illustrative example is given to verify that the $\Phi$-dependent dwell time technique is more effective than using mode-dependent dwell time [22] or dwell time.

The structure of this paper is as follows. Section 2 gives a description of the system and the necessary definitions and lemmas. Section 3 presents the main conclusions with their proofs. An illustrative example is given in Section 4. Section 5 provides the conclusion of this paper.

The notations used in this paper are shown in Table 1.

**Table 1.** The notations used in this paper.

| | |
|---|---|
| $\mathbb{R}$ | The set of real numbers |
| $\mathbb{R}^n$ | The set of $n$-dimensional real vectors |
| $\mathbb{R}^{n \times n}$ | The space of $n \times n$ real matrices |
| $\mathbb{N}\ (\mathbb{N}_+)$ | The set of nonnegative (positive) integers |
| $\mathfrak{G}^T$ | The transpose of a matrix $\mathfrak{G}$ |
| $\iota \succeq 0 (\iota \succ 0)$ | Each component of vector $\iota$ is nonnegative (positive) |
| $\underline{\delta}(\iota)(\overline{\delta}(\iota))$ | The minimal (maximal) components of vector $\iota$ |
| $\mathfrak{G} \succeq 0 (\mathfrak{G} \succ 0)$ | Each component of matrix $\mathfrak{G}$ is nonnegative (positive) |
| $\|x\|_1$ | 1-norm of $x(t)$; i.e., $\|x\|_1 = \sum_{i=1}^n \|x_i\|$ |
| $\|x\|_2$ | Euclidean vector norm of $x(t)$; i.e., $\|x\|_2 = (\sum_{i=1}^n x_i^2)^{\frac{1}{2}}$ |
| $\Longleftrightarrow$ | If and only if |

## 2. Problem Formulation and Preliminaries

Consider the continuous-time switched linear system

$$\dot{x}(s) = \mathfrak{G}_{\rho(s)} x(s), \quad s \in \mathbb{R}, \quad s \geq s_0, \tag{1}$$

and the discrete-time switched linear system

$$x(s+1) = \mathfrak{G}_{\rho(s)} x(s), \quad s \in \mathbb{N}, \quad s \geq s_0, \tag{2}$$

where $x(s) \in \mathbb{R}^n$ is the state of the system, and $\rho(s)$ is called a switching signal. $\rho(s)$ is a piece-wise constant right-continuous function: $[0, \infty) \to \mathfrak{M} = \{1, 2, \cdots, m\}$, where $m \in \mathbb{N}_+$

is the number of subsystems. $\mathfrak{G}_u \in \mathbb{R}^{n \times n}$ and $u \in \mathfrak{M}$ are known constant matrices of appropriate dimensions.

All subsystems studied in this paper are unstable. The switching instants have the following relationship: $0 \le s_0 < s_1 < \cdots < s_r < s_{r+1} < \cdots$, where $s_0$ represents the initial time of system operation and $s_r$ represents the $r$th switching instant. In addition, we assume that $x(s_r^+) = x(s_r^-)$, $r = 0, 1, 2, \cdots$. When $s \in [s_r, s_{r+1})$, we say the $\rho(s_r)$th subsystem of the switched system is active. The length of time between adjacent switching instants is called the dwell time $\tau_r = s_{r+1} - s_r, r = 0, 1, 2, \cdots$. Let $\mathfrak{K} = \{1, 2, \cdots, k\}$ where $k \in \mathbb{N}$ and $k \le m$. Define the surjection operator: $\Phi : \mathfrak{M} \mapsto \mathfrak{K}$. Set $\Phi_i = \{u \in \mathfrak{M} \mid \Phi(u) = i\}$ [23]. This work supposes that each family of $\Phi_i$ subsystems has a dwell time, denoted as $\tau_{\Phi_i, r}$, $r = 0, 1, 2, \cdots, i \in \mathfrak{K}$ and called the $\Phi$-dependent dwell time. If $\tau_r$ is too large, then the total system will be unstable due to it running unstable subsystems for a long time. If $\tau_r$ is too small, then overly fast switching also makes the system unstable. Thus, we limit the $\Phi$-dependent dwell time to a range that ensures the asymptotic stability of the system; namely, $\tau_{\Phi_i, r} \in [\tau_{\Phi_i, \min}, 2\tau_{a\Phi_i, \max} - \tau_{\Phi_i, \min}], i \in \mathfrak{K}$, $r = 0, 1, 2, \cdots$, where $\tau_{\Phi_i, \min}$ represents the minimum dwell time of a family of $\Phi_i$ subsystems, $\tau_{a\Phi_i, \max}$ represents the maximum average dwell time of a family of $\Phi_i$ subsystems, and $0 < \tau_{\Phi_i, \min} \le \tau_{a\Phi_i, \max}$. $\mathfrak{D}_{[\tau_{\Phi_i, \min}, 2\tau_{a\Phi_i, \max} - \tau_{\Phi_i, \min}]}$ is called the switching strategy set of the $\Phi$-dependent dwell time.

The aim of this study was to analyze the problem of the stability and robust stability of SPLSs where all subsystems are unstable. Before that, we first provide some definitions and lemmas to be used.

**Definition 1** ([24]). *Systems (1) and (2) are said to be positive if $x(s) \succeq 0, \forall s > s_0, \forall \rho(s)$, $x(s_0) \succeq 0$.*

**Definition 2** ([12]). *A matrix $\mathfrak{G}$ is said to be a Metzler matrix if its non-diagonal elements are positive or zero.*

Consider the continuous-time system

$$\dot{x}(s) = \mathfrak{G}x(s), \quad s \in \mathbb{R}, \quad s \ge s_0, \tag{3}$$

and the discrete-time system

$$x(s+1) = \mathfrak{G}x(s), \quad s \in \mathbb{N}, \quad s \ge s_0. \tag{4}$$

We note the following lemmas.

**Lemma 1** ([19]). *System (3) is positive $\iff \mathfrak{G}$ is a Metzler matrix, and System (4) is positive $\iff \mathfrak{G} \succeq 0$.*

From Lemma 1, we can deduce that System (1) is positive with regard to $\rho(s) \iff \mathfrak{G}_u \in \mathbb{R}^{n \times n}$ and $u \in \mathfrak{M}$ are Metzler matrices, and System (2) is positive with regard to $\rho(s) \iff \mathfrak{G}_u \succeq 0, u \in \mathfrak{M}$.

**Lemma 2** ([19]). *Let Systems (3) and (4) be positive; then, Systems (3) and (4) is asymptotically stable $\iff$ there exists a vector $\iota \succ 0$ such that $\mathfrak{G}^T \iota \prec 0$ ($(\mathfrak{G} - I)^T \iota \prec 0$).*

The function $\mathcal{F}(s) = x^T(s)\iota$ is said to be a linear co-positive Lyapunov function for the Systems (3) and (4) if there exists a vector $\iota \succ 0$ such that $\mathfrak{G}^T \iota \prec 0$ ($(\mathfrak{G} - I)^T \iota \prec 0$).

## 3. Main Results

This section presents sufficient conditions for the stability and robust stability of SPLSs in which all subsystems are unstable in the continuous-time and discrete-time cases.

### 3.1. Continuous-Time Case

In this section, sufficient conditions for the stability and robust stability of continuous SPLSs and their proofs are given.

**Theorem 1.** *Consider SPLS (1). Given scalars $\lambda_i > 0, 0 < \mu_i < 1, i \in \mathfrak{K}, 0 < \tau_{\Phi_i,\min} \leq \tau_{a\Phi_i,\max}$, if there exists a set of vectors $\iota_{u,h} \succ 0, h = 0, 1, 2, \cdots, \mathcal{H}, u \in \mathfrak{M}$ such that $\forall h = 0, 1, 2, \cdots, \mathcal{H}$, $\forall u, v \in \mathfrak{M}, i \in \mathfrak{K}$,*

$$\Lambda_{u,h}^T + \iota_{u,h}^T \mathfrak{G}_u - \lambda_i \iota_{u,h}^T \prec 0, \tag{5}$$

$$\Lambda_{u,h}^T + \iota_{u,h+1}^T \mathfrak{G}_u - \lambda_i \iota_{u,h+1}^T \prec 0, \tag{6}$$

$$\iota_{u,\mathcal{H}}^T \mathfrak{G}_u - \lambda_i \iota_{u,\mathcal{H}}^T \prec 0, \tag{7}$$

$$\iota_{v,0} - \mu_i \iota_{u,\mathcal{H}} \preceq 0, \tag{8}$$

$$\ln \mu_i + \lambda_i \tau_{a\Phi_i,\max} < 0, \tag{9}$$

*where $\Lambda_{u,h} = \frac{\mathcal{H}(\iota_{u,h+1} - \iota_{u,h})}{\tau_{\Phi_i,\min}}$, $\Phi(u) = i$, then System (1) is globally asymptotically stable under any switching law $\rho(s) \in \mathfrak{D}_{[\tau_{\Phi_i,\min}, 2\tau_{a\Phi_i,\max} - \tau_{\Phi_i,\min}]}$.*

**Analysis**. The Lyapunov functions constructed in most previous articles that were continuous in the dwell-time interval are not applicable when all subsystems are unstable. Using another method, we consider the construction of a discretized Lyapunov function to break the commutativity of Lyapunov functions in adjacent dwell-time intervals. In this way, systems for which all subsystems are unstable can be stabilized by proper switching.

**Proof.** Step 1: Prove that System (1) is stable.

For the convenience of narration, assume that $\rho(s_r) = u$, $\rho(s_{r+1}) = v$. Divide the interval $[s_r, s_r + \tau_{\Phi_i,\min})$ into $\mathcal{H}$ equal parts, with each interval represented as $I_{r,h}^u = [s_r + \omega_h^u, s_r + \omega_{h+1}^u)$, $h = 0, 1, \cdots, \mathcal{H} - 1$. The length of the equal division is $l_u = \frac{\tau_{\Phi_i,\min}}{\mathcal{H}}$, where $\omega_h^u = h \cdot l_u, h = 0, 1, \cdots, \mathcal{H}$.

We use the linear interpolation formula to construct linear functions for each segment of the minimum dwell-time interval. When $s \in I_{r,h}^u$, let $\iota_u(s) = (1 - \alpha)\iota_{u,h} + \alpha\iota_{u,h+1}$, where $\alpha = \frac{s - s_r - \omega_h^u}{l_u}$. Substituting the expressions of $\alpha$ and $l_u$ into the above equation yields $\dot{\iota}_u(s) = \Lambda_{u,h}$. When $\alpha = 0$, we denote $\iota_u(s_r + \omega_h^u)$ as $\iota_{u,h}$.

When $s \in [s_r + \tau_{\Phi_i,\min}, s_{r+1})$, let $\iota_u(s)$ remain the value of the left endpoint during this period; i.e., $\iota_u(s) = \iota_{u,\mathcal{H}}$. Obviously, $\dot{\iota}_u(s) = 0$.

Since the system we are studying is positive, we can construct the following multiple co-positive Lyapunov functions:

$$\mathcal{F}_u(s) = \iota_u^T(s)x(s), u \in \mathfrak{M}. \tag{10}$$

Among them, the vector $\iota_u(s)$ is as defined above.

When $s \in I_{r,h}^u$,

$$\begin{aligned} \dot{\mathcal{F}}_u(s) &= \Lambda_{u,h}^T x(s) + \iota_u^T(s)\mathfrak{G}_u x(s) \\ &= \Lambda_{u,h}^T x(s) + [(1 - \alpha)\iota_{u,h}^T + \alpha\iota_{u,h+1}^T]\mathfrak{G}_u x(s) \\ &= (1 - \alpha)[\Lambda_{u,h}^T + \iota_{u,h}^T\mathfrak{G}_u]x(s) + \alpha[\Lambda_{u,h}^T + \iota_{u,h+1}^T\mathfrak{G}_u]x(s). \end{aligned} \tag{11}$$

Combining Equations (5) and (6), it can be concluded that $\dot{\mathcal{F}}_u(s) < \lambda_i \mathcal{F}_u(s)$.

When $s \in [s_r + \tau_{\Phi_i,\min}, s_{r+1})$, $\dot{\mathcal{F}}_u(s) = \iota_{u,\mathcal{H}}^T\mathfrak{G}_u x(s)$. Combining this with Equation (7), it can be concluded that $\dot{\mathcal{F}}_u(s) < \lambda_i \mathcal{F}_u(s)$.

From the above analysis, it can be seen that

$$\dot{\mathcal{F}}_u(s) < \lambda_i \mathcal{F}_u(s), s \in [s_r, s_{r+1}). \tag{12}$$

According to Equation (8) and the assumption that the system state does not jump during switching, it can be concluded that

$$\mathcal{F}_v(s_{r+1}^+) \leq \mu_i \mathcal{F}_u(s_{r+1}^-), \ u \neq v, \forall u, v \in \mathfrak{M}. \tag{13}$$

Combining Equations (12) and (13), we have

$$
\begin{aligned}
\mathcal{F}_{\rho(s_r)}(s) &< e^{\lambda_{\Phi(\rho(s_r))}(s-s_r)} \mathcal{F}_{\rho(s_r)}(s_r) \\
&\leq e^{\lambda_{\Phi(\rho(s_r))}(s-s_r)} \mu_{\Phi(\rho(s_{r-1}))} \mathcal{F}_{\rho(s_{r-1})}(s_r^-) \\
&< e^{\lambda_{\Phi(\rho(s_r))}(s-s_r)} \mu_{\Phi(\rho(s_{r-1}))} e^{\lambda_{\Phi(\rho(s_{r-1}))}(s_r-s_{r-1})} \mathcal{F}_{\rho(s_{r-1})}(s_{r-1}) \\
&\vdots \\
&< e^{\lambda_{\Phi(\rho(s_r))}(s-s_r)} \mu_{\Phi(\rho(s_{r-1}))} \mu_{\Phi(\rho(s_{r-2}))} \cdots \mu_{\Phi(\rho(s_0))} \\
&\quad e^{\lambda_{\Phi(\rho(s_{r-1}))}(s_r-s_{r-1})} \cdots e^{\lambda_{\Phi(\rho(s_0))}(s_1-s_0)} \mathcal{F}_{\rho(s_0)}(s_0) \\
&= e^{\lambda_{\Phi(\rho(s_r))}(s-s_r)} \left( \prod_{i=1}^{k} \mu_i^{n_i} e^{\sum_{i=1}^{k} \lambda_i T_i} \right) \mathcal{F}_{\rho(s_0)}(s_0) \\
&\leq e^{\lambda_{\Phi(\rho(s_r))} \cdot 2\tau_{a\Phi_i,max}} \left( \prod_{i=1}^{k} \mu_i^{n_i} e^{\lambda_i n_i \tau_{a\Phi_i,max}} \right) \mathcal{F}_{\rho(s_0)}(s_0),
\end{aligned}
\tag{14}
$$

where $n_i$ denotes the switching numbers of the $i$ th family subsystems over the interval $[0, s]$, and $T_i$ denotes the total activated time of the $i$ th family subsystems over the interval $[0, s]$. Let $\tau = \max\{\tau_{a\Phi_i,max}\}, \lambda = \max\{\lambda_i\}, i \in \mathfrak{K}$. From Equation (9), we can derive $\mu_i e^{\lambda_i \tau_{a\Phi_i,max}} < 1$; namely, $\mu_i^{n_i} e^{\lambda_i n_i \tau_{a\Phi_i,max}} < 1, \forall i \in \mathfrak{K}$. Substituting into Equation (14), we obtain:

$$\mathcal{F}_{\rho(s_r)}(s) < e^{2\lambda\tau} \mathcal{F}_{\rho(s_0)}(s_0). \tag{15}$$

As we all know, the following inequalities are true: $\mathcal{F}_{\rho(s_0)}(s_0) \leq \overline{\delta}\sqrt{n}\|x(s_0)\|_2$, $\underline{\delta}\|x(s)\|_2 \leq \mathcal{F}_{\rho(s_r)}(s)$, where $\overline{\delta} = \max\limits_{u \in \mathfrak{M}, h=0,1,\cdots,\mathcal{H}} \{\overline{\delta}(\iota_{u,h})\}, \underline{\delta} = \min\limits_{u \in \mathfrak{M}, h=0,1,\cdots,\mathcal{H}} \{\underline{\delta}(\iota_{u,h})\}$. Substituting into Equation (15), we obtain:

$$\|x(s)\|_2 \leq \Theta\|x(s_0)\|_2, \tag{16}$$

where $\Theta = \frac{\overline{\delta}\sqrt{n}}{\underline{\delta}} e^{2\lambda\tau}$. Then, $\forall \epsilon > 0$, we can choose $\|x(s_0)\|_2 < \xi(\epsilon) = \Theta^{-1}\epsilon$. This results in $\|x(s)\|_2 < \epsilon$, and the stability of System (1) can be obtained.

Step 2: Prove that System (1) is globally asymptotically stable.

Combining Equations (12) and (13), it can be seen that $\mathcal{F}_v(s_{r+1}) < \gamma\mathcal{F}_u(s_r), 0 < \gamma < 1$, where $\gamma = \max(\mu_i e^{2\lambda_i \tau_{a\Phi_i,max}})$. Recursively, the following relationship can be obtained: $\mathcal{F}_{\rho(s_r)}(s_r) < \gamma^r \mathcal{F}_{\rho(s_0)}(s_0)$. We can derive $\lim\limits_{r\to\infty} \mathcal{F}_{\rho(s_r)}(s_r) = 0$, which implies $\lim\limits_{r\to\infty} x(s_r) = 0$.

Below, we use the proof by contradiction to obtain the desired conclusion. Assume the existence of $x(s_f)$ such that $\lim\limits_{f\to\infty} x(s_f) = c$, where $c$ is a positive constant. From the definition of the limit, it can be seen that there exists $q > 0$ such that $\|x(s_f)\|_2 > c - a$ whenever $f > q$, where $a(< c)$ is any positive constant.

From the stability of System (1), it can be seen that there exists $\zeta(p) > 0$ such that $\|x(s)\|_2 < p$ whenever $\|x(s_0)\| < \zeta$, where $p$ is positive. Let $p = c - a$; then, $\|x(s)\|_2 < c - a$. This contradicts $\|x(s_f)\|_2 > c - a$, so the assumption is not valid, Then, we have $\lim\limits_{s\to\infty} x(s) = 0$.

Hence, SPLS (1) is globally asymptotically stable with regard to $\rho(s) \in \mathfrak{D}_{[\tau_{\Phi_i,min}, 2\tau_{a\Phi_i,max}-\tau_{\Phi_i,min}]}$. $\square$

**Remark 1.** *The co-positive Lyapunov functions in Theorem 1 are time-varying during $[s_r, s_r + \tau_{\Phi_i,\min})$ and time-invariant during $[s_r + \tau_{\Phi_i,\min}, s_{r+1})$, where $r = 0, 1, \cdots$. The discretized co-positive Lyapunov function divides the dwell-time interval into a finite number of small regions, and the vector function $\iota_u(s)$ varies linearly with respect to $s$ in each small region.*

**Remark 2.** *The Lyapunov functions $\mathcal{F}_u(s)$ in Theorem 1 are allowed to increase while the unstable subsystem is working, and the increase rate is limited to $\dot{\mathcal{F}}_u(s) < \lambda_i \mathcal{F}_u(s)$. Moreover, this increment can be compensated for by the switching behavior; that is, the Lyapunov functions do not increase at the switching instants.*

**Remark 3.** *The switching strategy set can be obtained by the following procedure. Firstly, appropriate $\lambda_i$, $\mu_i$, $\mathcal{H}$, and $\tau_{\Phi_i,\min}$ are given according to the constraints of each parameter. Then, the values of vectors $\iota_{u,h}$ are calculated by linear programming. We can then obtain the switching strategy set. Later, we will visually see through examples that the $\Phi$-dependent dwell time technique is more flexible than the mode-dependent dwell time and the dwell time.*

**Remark 4.** *Unlike most stability results based on various dwell time strategies or their extended forms, Theorem 1 applies to the situation where all subsystems are unstable. The theorem is based on the fact that the energy increment caused by unstable subsystem may be compensated for by the suitable "stable" switching. This has two limitations: the computational complexity caused by group diversity and the blindness in determining the minimum dwell time. The conditions of the theorem are to some extent harsh, but it provides an effective switching design scheme for related research.*

Due to the fact that real-world systems are always influenced by various factors, the following investigates the robust stability of continuous switched-interval positive systems where all subsystems are unstable.

Consider the continuous switched-interval positive system

$$\dot{x}(s) = \mathfrak{G}_{\rho(s)} x(s), \tag{17}$$

where $\mathfrak{G}_u \in [\underline{\mathfrak{G}}_u, \overline{\mathfrak{G}}_u], u \in \mathfrak{M}$, $\underline{\mathfrak{G}}_u$, and $\overline{\mathfrak{G}}_u, u \in \mathfrak{M}$ are constant matrices with suitable dimensions, representing interval uncertainty. In addition, we assume that $\underline{\mathfrak{G}}_u$ are Metzler matrices.

**Theorem 2.** *Consider SPLS (17). Given scalars $\lambda_i > 0$, $0 < \mu_i < 1$, $i \in \mathfrak{K}$, and $0 < \tau_{\Phi_i,\min} \leq \tau_{a\Phi_i,\max}$, if there exists a set of vectors $\iota_{u,h} \succ 0, h = 0, 1, 2, \cdots, \mathcal{H}, u \in \mathfrak{M}$ such that $\forall h = 0, 1, 2, \cdots, \mathcal{H}, \forall u, v \in \mathfrak{M}, i \in \mathfrak{K}$,*

$$\Lambda_{u,h}^T + \iota_{u,h}^T \overline{\mathfrak{G}}_u - \lambda_i \iota_{u,h}^T \prec 0, \tag{18}$$

$$\Lambda_{u,h}^T + \iota_{u,h+1}^T \overline{\mathfrak{G}}_u - \lambda_i \iota_{u,h+1}^T \prec 0, \tag{19}$$

$$\iota_{u,\mathcal{H}}^T \overline{\mathfrak{G}}_u - \lambda_i \iota_{u,\mathcal{H}}^T \prec 0, \tag{20}$$

$$\iota_{v,0} - \mu_i \iota_{u,\mathcal{H}} \preceq 0, \tag{21}$$

$$\ln \mu_i + \lambda_i \tau_{a\Phi_i,\max} < 0, \tag{22}$$

*where $\Lambda_{u,h} = \frac{\mathcal{H}(\iota_{u,h+1} - \iota_{u,h})}{\tau_{\Phi_i,\min}}$, $\Phi(u) = i$, then System (17) is globally asymptotically stable under any switching law $\rho(s) \in \mathfrak{D}_{[\tau_{\Phi_i,\min}, 2\tau_{a\Phi_i,\max} - \tau_{\Phi_i,\min}]}$.*

**Proof.** Under the assumption that $\underline{\mathfrak{G}}_u$ are Metzler matrices, according to Definition 2, it can be concluded that $\mathfrak{G}_u$ are also Metzler matrices. We still use the discretized co-positive Lyapunov function (Equation (10)).

When $s \in I_{r,h}^u$,

$$
\begin{aligned}
\dot{\mathcal{F}}_u(s) &\leq \Lambda_{u,h}^T x(s) + \iota_u^T(s)\overline{\mathfrak{G}}_u x(s) \\
&= (1-\alpha)[\Lambda_{u,h}^T + \iota_{u,h}^T \overline{\mathfrak{G}}_u]x(s) + \alpha[\Lambda_{u,h}^T + \iota_{u,h+1}^T \overline{\mathfrak{G}}_u]x(s).
\end{aligned}
\tag{23}
$$

Combining Equations (18) and (19), it can be concluded that $\dot{\mathcal{F}}_u(s) < \lambda_i \mathcal{F}_u(s)$. Similarly, when $s \in [s_r + \tau_{\Phi_i,\min}, s_{r+1})$, there is the same formula. In summary, it can be concluded that $\dot{\mathcal{F}}_u(s) < \lambda_i \mathcal{F}_u(s), s \in [s_r, s_{r+1})$. The subsequent proof is similar to Theorem 1. Therefore, there will be no further explanation. $\square$

*3.2. Discrete-Time Case*

This section provides the sufficient conditions for the stability and robust stability of discrete-time SPLSs.

**Theorem 3.** *Consider SPLS* (2). *Given scalars* $\lambda_i > 0, 0 < \mu_i < 1, i \in \mathfrak{K}$, *and* $0 < \tau_{\Phi_i,\min} \leq \tau_{a\Phi_i,\max}$, *if there exists a set of vectors* $\iota_{u,h} \succ 0, h = 0, 1, 2, \cdots, \mathcal{H}, u \in \mathfrak{M}$ *such that* $\forall h = 0, 1, 2, \cdots, \mathcal{H}, \forall u, v \in \mathfrak{M}, i \in \mathfrak{K}$,

$$
\Lambda_{u,h}^T \mathfrak{G}_u + \iota_{u,h}^T(\mathfrak{G}_u - I) - \lambda_i \iota_{u,h}^T \prec 0,
\tag{24}
$$

$$
\Lambda_{u,h}^T \mathfrak{G}_u + \iota_{u,h+1}^T(\mathfrak{G}_u - I) - \lambda_i \iota_{u,h+1}^T \prec 0,
\tag{25}
$$

$$
\iota_{u,\mathcal{H}}^T(\mathfrak{G}_u - I) - \lambda_i \iota_{u,\mathcal{H}}^T \prec 0,
\tag{26}
$$

$$
\iota_{v,0} - \mu_i \iota_{u,\mathcal{H}} \preceq 0,
\tag{27}
$$

$$
\ln \mu_i + \tau_{a\Phi_i,\max} \ln(1 + \lambda_i) < 0,
\tag{28}
$$

*where* $\Lambda_{u,h} = \frac{\mathcal{H}(\iota_{u,h+1} - \iota_{u,h})}{\tau_{\Phi_i,\min}}$, $\Phi(u) = i$, *then System* (2) *is globally asymptotically stable under any switching law* $\rho(s) \in \mathfrak{D}_{[\tau_{\Phi_i,\min}, 2\tau_{a\Phi_i,\max} - \tau_{\Phi_i,\min}]}$.

**Proof.** We still use the linear interpolation formula to construct linear functions for each segment of the minimum dwell-time interval $I_{r,h}^u = [s_r + \omega_h^u, s_r + \omega_{h+1}^u), h = 0, 1, \cdots, \mathcal{H} - 1$, as shown in Theorem 1. We can easily obtain $\iota_u(s+1) = \Lambda_{u,h} + \iota_u(s)$. Then,

$$
\begin{aligned}
\Delta \mathcal{F}_u(s) &= \mathcal{F}_u(s+1) - \mathcal{F}_u(s) \\
&= (\iota_u^T(s) + \Lambda_{u,h}^T)\mathfrak{G}_u x(s) - \iota_u^T(s)x(s) \\
&= [(1-\alpha)\iota_{u,h}^T + \alpha\iota_{u,h+1}^T + \Lambda_{u,h}^T]\mathfrak{G}_u x(s) - [(1-\alpha)\iota_{u,h}^T + \alpha\iota_{u,h+1}^T]x(s) \\
&= (1-\alpha)[\iota_{u,h}^T(\mathfrak{G}_u - I) + \Lambda_{u,h}^T \mathfrak{G}_u]x(s) \\
&\quad + \alpha[\iota_{u,h+1}^T(\mathfrak{G}_u - I) + \Lambda_{u,h}^T \mathfrak{G}_u]x(s).
\end{aligned}
\tag{29}
$$

Combining Equations (24) and (25), it can be concluded that $\mathcal{F}_u(s+1) < (1 + \lambda_i)\mathcal{F}_u(s)$.

Similarly, when $s \in [s_r + \tau_{\Phi_i,\min}, s_{r+1})$, take $\iota_u(s) = \iota_{u,\mathcal{H}}$. Using Equation (26), we can derive

$$
\begin{aligned}
\Delta \mathcal{F}_u(s) &= \iota_{u,\mathcal{H}}^T \mathfrak{G}_u x(s) - \iota_{u,\mathcal{H}}^T x(s) \\
&= \iota_{u,\mathcal{H}}^T(\mathfrak{G}_u - I)x(s) \\
&< \lambda_i \mathcal{F}_u(s).
\end{aligned}
\tag{30}
$$

From the above analysis, it can be seen that

$$
\mathcal{F}_u(s+1) < (1 + \lambda_i)\mathcal{F}_u(s), s \in [s_r, s_{r+1}).
\tag{31}
$$

According to Equation (27) and the assumption that the system state does not jump during switching, it can be concluded that

$$\mathcal{F}_v(s_{r+1}^+) \leq \mu_i \mathcal{F}_u(s_{r+1}^-), \; u \neq v, \; \forall u, v \in \mathfrak{M}. \tag{32}$$

From Equations (31) and (32), we can obtain

$$
\begin{aligned}
\mathcal{F}_{\rho(s_r)}(s) &< (1 + \lambda_{\Phi(\rho(s_r))})^{s-s_r} \mathcal{F}_{\rho(s_r)}(s_r) \\
&\leq (1 + \lambda_{\Phi(\rho(s_r))})^{s-s_r} \mu_{\Phi(\rho(s_{r-1}))} \mathcal{F}_{\rho(s_{r-1})}(s_r) \\
&< (1 + \lambda_{\Phi(\rho(s_r))})^{s-s_r} \mu_{\Phi(\rho(s_{r-1}))} (1 + \lambda_{\Phi(\rho(s_{r-1}))})^{s_r - s_{r-1}} \mathcal{F}_{\rho(s_{r-1})}(s_{r-1}) \\
&\vdots \\
&< (1 + \lambda_{\Phi(\rho(s_r))})^{s-s_r} \mu_{\Phi(\rho(s_{r-1}))} \mu_{\Phi(\rho(s_{r-2}))} \cdots \mu_{\Phi(\rho(s_0))} \\
&\quad (1 + \lambda_{\Phi(\rho(s_{r-1}))})^{s_r - s_{r-1}} \cdots (1 + \lambda_{\Phi(\rho(s_0))})^{s_1 - s_0} \mathcal{F}_{\rho(s_0)}(s_0) \\
&= (1 + \lambda_{\Phi(\rho(s_r))})^{s-s_r} \left( \prod_{i=1}^k \mu_i^{n_i} (1 + \lambda_i)^{T_i} \right) \mathcal{F}_{\rho(s_0)}(s_0) \\
&\leq (1 + \lambda_{\Phi(\rho(s_r))})^{2\tau_{a\Phi_i,\max}} \left( \prod_{i=1}^k \mu_i^{n_i} (1 + \lambda_i)^{n_i \tau_{a\Phi_i,\max}} \right) \mathcal{F}_{\rho(s_0)}(s_0),
\end{aligned}
\tag{33}
$$

where $n_i$ denotes the switching numbers of the $i$ th family subsystems over the interval $[0, s]$, and $T_i$ denotes the total activated time of the $i$ th family subsystems over the interval $[0, s]$. Let $\Psi = \max\{(1 + \lambda_i)^{2\tau_{a\Phi_i,\max}}\}$. From Equation (28), we can derive $\mu_i(1 + \lambda_i)^{\tau_{a\Phi_i,\max}} < 1$. Substituting into Equation (33), we obtain:

$$\mathcal{F}_{\rho(s_r)}(s) < \Psi \mathcal{F}_{\rho(s_0)}(s_0). \tag{34}$$

As we all know, the following inequalities are true: $\mathcal{F}_{\rho(s_0)}(s_0) \leq \overline{\delta}\sqrt{n}\|x(s_0)\|_2$, $\underline{\delta}\|x(s)\|_2 \leq \mathcal{F}_{\rho(s_r)}(s)$, where $\overline{\delta} = \max\limits_{u \in \mathfrak{M}, h=0,1,\cdots,\mathcal{H}} \{\overline{\delta}(\iota_{u,h})\}$, $\underline{\delta} = \min\limits_{u \in \mathfrak{M}, h=0,1,\cdots,\mathcal{H}} \{\underline{\delta}(\iota_{u,h})\}$. Substituting into Equation (34), we obtain:

$$\|x(s)\|_2 \leq \Omega\|x(s_0)\|_2, \tag{35}$$

where $\Omega = \frac{\overline{\delta}\sqrt{n}}{\underline{\delta}}\Psi$. Then, $\forall \epsilon > 0$, and we can choose $\|x(s_0)\|_2 < \xi(\epsilon) = \Omega^{-1}\epsilon$. This results in $\|x(s)\|_2 < \epsilon$, and the stability of System (2) can be obtained.

The rest of the proof of Theorem 3 is similar to Theorem 1 and is therefore omitted. $\square$

Below, we provide sufficient conditions for the case of interval uncertainty.
Consider the discrete switched-interval positive system

$$x(s+1) = \mathfrak{G}_{\rho(s)}x(s), \tag{36}$$

where $\mathfrak{G}_u \in [\underline{\mathfrak{G}}_u, \overline{\mathfrak{G}}_u], u \in \mathfrak{M}, \underline{\mathfrak{G}}_u$, and $\overline{\mathfrak{G}}_u, u \in \mathfrak{M}$ are the same as Equation (17). In addition, we assume that $\underline{\mathfrak{G}}_u \succeq 0$.

**Theorem 4.** *Consider Equation (36). Given scalars $\lambda_i > 0, 0 < \mu_i < 1, i \in \mathfrak{K}$, and $0 < \tau_{\Phi_i,\min} \leq \tau_{a\Phi_i,\max}$, if there exists a set of vectors $\iota_{u,h} \succ 0, h = 0,1,2,\cdots,\mathcal{H}, u \in \mathfrak{M}$ such that $\forall h = 0,1,2,\cdots,\mathcal{H}, \forall u, v \in \mathfrak{M}, i \in \mathfrak{K}$,*

$$\Lambda_{u,h}^T \overline{\mathfrak{G}}_u + \iota_{u,h}^T(\overline{\mathfrak{G}}_u - I) - \lambda_i \iota_{u,h}^T \prec 0, \tag{37}$$

$$\Lambda_{u,h}^T \overline{\mathfrak{G}}_u + \iota_{u,h+1}^T(\overline{\mathfrak{G}}_u - I) - \lambda_i \iota_{u,h+1}^T \prec 0, \tag{38}$$

$$\iota_{u,\mathcal{H}}^T(\overline{\mathfrak{G}}_u - I) - \lambda_i \iota_{u,\mathcal{H}}^T \prec 0, \tag{39}$$

$$\iota_{v,0} - \mu_i \iota_{u,\mathcal{H}} \preceq 0, \tag{40}$$

$$\ln \mu_i + \tau_{a\Phi_i,\max} \ln(1 + \lambda_i) < 0, \tag{41}$$

where $\Lambda_{u,h} = \frac{\mathcal{H}(\iota_{u,h+1} - \iota_{u,h})}{\tau_{\Phi_i,\min}}$ and $\Phi(u) = i$; then, System (36) is globally asymptotically stable under any switching law $\rho(s) \in \mathfrak{D}_{[\tau_{\Phi_i,\min}, 2\tau_{a\Phi_i,\max} - \tau_{\Phi_i,\min}]}$.

The proof of this theorem is basically the same as the previous proof and is omitted here.

## 4. Illustrative Example

In this section, we illustrate through a specific example that the $\Phi$-dependent dwell time technique is more flexible than mode-dependent dwell time and dwell time.

Consider an epidemiological model with $n$ population groups [25]. Every $\iota$ th group is divided into infectives and susceptibles. Let $I_\iota(s)$ and $S_\iota(s)$, respectively, denote the number of infectives and susceptibles at time $s$. Suppose the total number $I_\iota(s) + S_\iota(s) = N_\iota$ is constant at any time $s \geq 0$. By taking $x_\iota(s) = I_\iota(s) / N_\iota$, we have, for $\iota = 1, 2, \cdots, n$:

$$\dot{x}_\iota(s) = (1 - x_\iota(s)) \sum_{j=1}^{N} \frac{\beta_{\iota j} N_j}{N_\iota} x_j(s) - (\eta_\iota + \omega_\iota) x_\iota(s), \tag{42}$$

where $\beta_{\iota j} > 0$ and $\eta_\iota > 0$ are some known constants, and $\omega_\iota > 0$ is the death rate in the $\iota$ group. Suppose there are $m$ different therapies to fight the epidemic, and the infection rate $\eta_{\rho(s)}$ is not constant but depends, at each time $s$, on $\rho(s) \in \{1, 2, \cdots, m\}$, which orchestrates the different therapies for the different population groups. By linearizing the system around the disease-free point $x = 0$, one can get the following system:

$$\dot{x}(s) = \mathfrak{G}_{\rho(s)} x(s); \tag{43}$$

i.e., Equation (1) with the system's matrices given as follows:

$$\mathfrak{G}_1 = \begin{bmatrix} -2.75 & 0.05 \\ 0.045 & 0.15 \end{bmatrix}, \mathfrak{G}_2 = \begin{bmatrix} 0.1 & 0.035 \\ 0.03 & -1.75 \end{bmatrix}, \mathfrak{G}_3 = \begin{bmatrix} 0.2 & 0.06 \\ 0.05 & -3 \end{bmatrix}.$$

The eigenvalues of $\mathfrak{G}_1$ are $a_1 = -2.7508$ and $a_2 = 0.1508$, the eigenvalues of $\mathfrak{G}_2$ are $a_1 = 0.1006$ and $a_2 = -1.7506$, and the eigenvalues of $\mathfrak{G}_3$ are $a_1 = 0.2009$ and $a_2 = -3.0009$. Obviously, all three subsystems are unstable.

For the convenience of calculation, we take $\mathcal{H} = 1$, $x(0) = [3, 5]^T$. $\mathfrak{D}_i$ represents the switching strategy set of the $i$ th family subsystem, and $\tau_i$ represents the dwell time of the $i$ th family subsystem. Due to the number of subsystems being three, the set $\mathfrak{K}$ can be divided into the following three situations. According to the choice of the different $\Phi$, one can get different switching strategies. When $\mathfrak{K} = \{1\}$, then $\Phi_1 = \{1, 2, 3\}$, which corresponds to the case with the dwell time technique. When $\mathfrak{K} = \{1, 2\}$, there are three types of grouping: (1) $\Phi_1 = \{1, 2\}$, $\Phi_2 = \{3\}$; (2) $\Phi_1 = \{1, 3\}$, $\Phi_2 = \{2\}$; and (3) $\Phi_1 = \{1\}$, $\Phi_2 = \{2, 3\}$. When $\mathfrak{K} = \{1, 2, 3\}$, then $\Phi_1 = \{1\}$, $\Phi_2 = \{2\}$, and $\Phi_3 = \{3\}$, which corresponds to the mode-dependent dwell time technique. In each group, select appropriate $\lambda_i > 0$, $0 < \mu_i < 1$, and $\tau_{\Phi_i,\min}$, and use linear programming to solve the vector $\iota_{u,h}$ that satisfies Equations (5)–(8) in Theorem 1. When Equations (5)–(8) have a solution, by bringing the values of $\lambda_i$ and $\mu_i$ into Equation (9), we can obtain $\tau_{a\Phi_i,\max}$. The switching strategy set $\mathfrak{D}_i$ can be obtained from $\tau_{\Phi_i,\min}$ and $\tau_{a\Phi_i,\max}$. Select the appropriate switching $\tau_i$ in $\mathfrak{D}_i$. Based on the selected $\tau_i$, the state response (Figures 1–5) can be obtained for different groups $\Phi$. The following table provides the parameter selection and switching strategy set corresponding to the dwell time, $\Phi$-dependent dwell time, and mode-dependent dwell time techniques, respectively. As can be seen from the state response figures, all three techniques can stabilize the system with appropriate switching. It can be seen from Table 2 that the $\tau_{\Phi_i,\min}$ of the dwell tine technique is 0.5, while the $\tau_{\Phi_i,\min}$ can be 0.4 in the three cases where $\mathfrak{K} = \{1, 2\}$. Similarly, the $\Phi$-dependent dwell time technique has a broader range of switching strategy sets than the mode-dependent one. By comparing the switching signal sets and state response graphs of the different techniques, it can be seen that the conclusions obtained with the

Φ-dependent dwell time cover the previous dwell time and mode-dependent dwell time. The technique used in this paper can make the system stable in the shortest time with the appropriate switching.

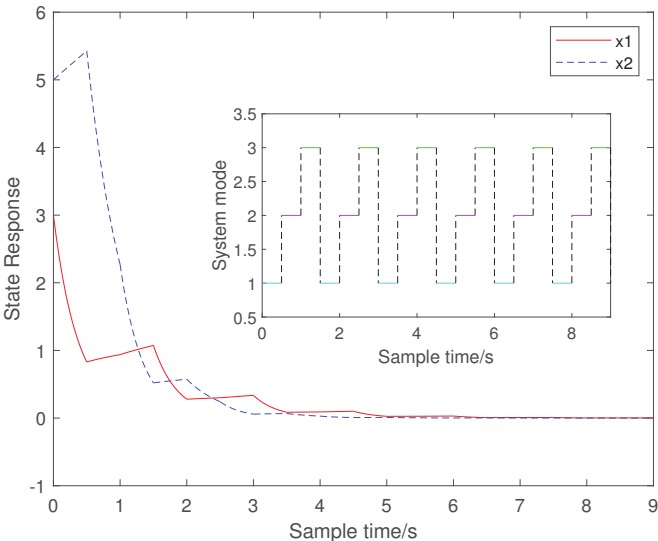

**Figure 1.** The state response of the system with the dwell time technique.

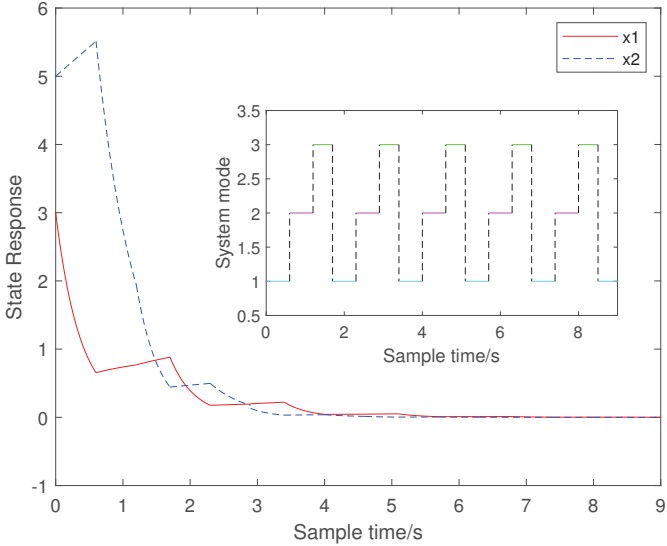

**Figure 2.** The state response of the system with group one of the Φ-dependent dwell time technique.

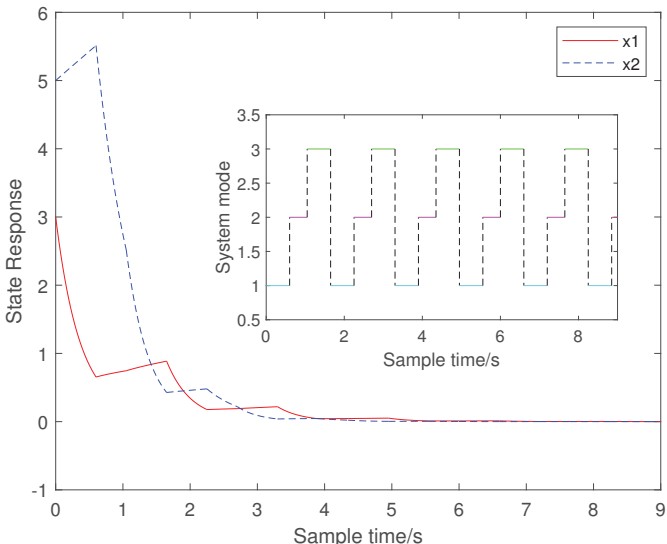

**Figure 3.** The state response of the system with group two of the Φ-dependent dwell time technique.

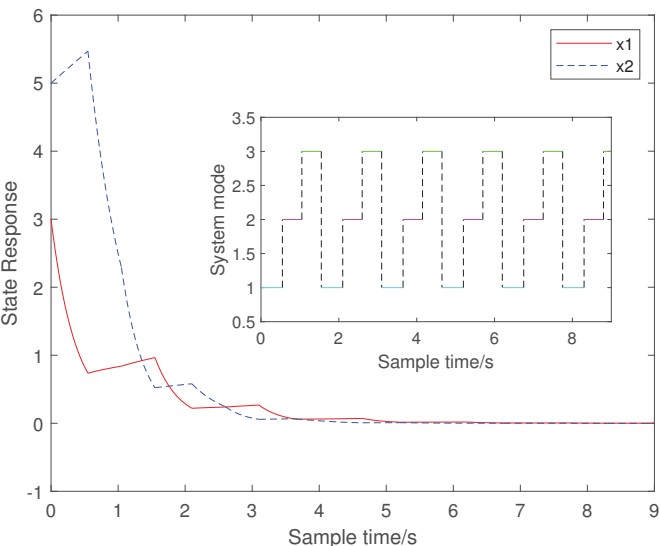

**Figure 4.** The state response of the system with group three of the Φ-dependent dwell time technique.

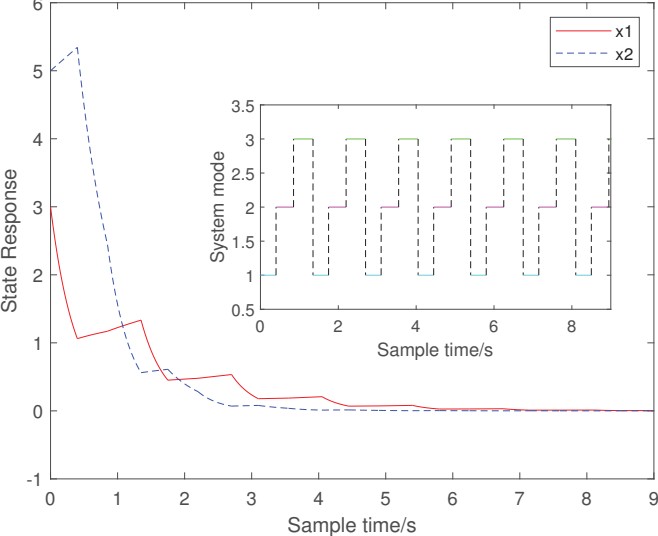

**Figure 5.** The state response of the system with the mode-dependent dwell time technique.

**Table 2.** Comparison of the three methods.

| Technique | Dwell Time $\mathfrak{K}=\{1\}$ | $\mathfrak{K}=\{1,2\}$ | | | Mode-Dependent Dwell Time $\mathfrak{K}=\{1,2,3\}$ |
|---|---|---|---|---|---|
| | | \multicolumn{3}{c}{**Φ-Dependent Dwell Time**} | |
| $\Phi$ | $\Phi_1=\{1,2,3\}$ | $\Phi_1=\{1,2\}$ $\Phi_2=\{3\}$ | $\Phi_1=\{1,3\}$ $\Phi_2=\{2\}$ | $\Phi_1=\{1\}$ $\Phi_2=\{2,3\}$ | $\Phi_1=\{1\}$ $\Phi_2=\{2\}$ $\Phi_3=\{3\}$ |
| $\lambda$ | $\lambda_1=0.5$ | $\lambda_1=0.5$ $\lambda_2=0.55$ | $\lambda_1=0.5$ $\lambda_2=0.57$ | $\lambda_1=0.5$ $\lambda_2=0.6$ | $\lambda_1=0.5$ $\lambda_2=0.6$ $\lambda_3=0.55$ |
| $\mu$ | $\mu_1=0.75$ | $\mu_1=0.7$ $\mu_2=0.65$ | $\mu_1=0.7$ $\mu_2=0.6$ | $\mu_1=0.7$ $\mu_2=0.68$ | $\mu_1=0.75$ $\mu_2=0.7$ $\mu_3=0.65$ |
| $l_{1,0}$ | $\begin{bmatrix}0.0100\\0.0116\end{bmatrix}$ | $\begin{bmatrix}0.0100\\0.0124\end{bmatrix}$ | $\begin{bmatrix}0.0100\\0.0124\end{bmatrix}$ | $\begin{bmatrix}0.0100\\0.0127\end{bmatrix}$ | $\begin{bmatrix}0.0100\\0.0119\end{bmatrix}$ |

**Table 2.** *Cont.*

| Technique | Dwell Time $\mathfrak{K}=\{1\}$ | $\mathfrak{K}=\{1,2\}$ | | | Mode-Dependent Dwell Time $\mathfrak{K}=\{1,2,3\}$ |
|---|---|---|---|---|---|
| | | \multicolumn{3}{c}{**Φ-Dependent Dwell Time**} | |
| $l_{1,1}$ | $\begin{bmatrix}0.0177\\0.0133\end{bmatrix}$ | $\begin{bmatrix}0.0234\\0.0143\end{bmatrix}$ | $\begin{bmatrix}0.0255\\0.0143\end{bmatrix}$ | $\begin{bmatrix}0.0226\\0.0143\end{bmatrix}$ | $\begin{bmatrix}0.0214\\0.0133\end{bmatrix}$ |
| $l_{2,0}$ | $\begin{bmatrix}0.0132\\0.0100\end{bmatrix}$ | $\begin{bmatrix}0.0164\\0.0100\end{bmatrix}$ | $\begin{bmatrix}0.0178\\0.0100\end{bmatrix}$ | $\begin{bmatrix}0.0158\\0.0100\end{bmatrix}$ | $\begin{bmatrix}0.0161\\0.0100\end{bmatrix}$ |
| $l_{2,1}$ | $\begin{bmatrix}0.0157\\0.0133\end{bmatrix}$ | $\begin{bmatrix}0.0195\\0.0143\end{bmatrix}$ | $\begin{bmatrix}0.0211\\0.0167\end{bmatrix}$ | $\begin{bmatrix}0.0189\\0.0147\end{bmatrix}$ | $\begin{bmatrix}0.0195\\0.0143\end{bmatrix}$ |
| $l_{3,0}$ | $\begin{bmatrix}0.0118\\0.0100\end{bmatrix}$ | $\begin{bmatrix}0.0137\\0.0100\end{bmatrix}$ | $\begin{bmatrix}0.0126\\0.0100\end{bmatrix}$ | $\begin{bmatrix}0.0128\\0.0100\end{bmatrix}$ | $\begin{bmatrix}0.0137\\0.0100\end{bmatrix}$ |
| $l_{3,1}$ | $\begin{bmatrix}0.0133\\0.0154\end{bmatrix}$ | $\begin{bmatrix}0.0154\\0.0190\end{bmatrix}$ | $\begin{bmatrix}0.0143\\0.0177\end{bmatrix}$ | $\begin{bmatrix}0.0147\\0.0187\end{bmatrix}$ | $\begin{bmatrix}0.0154\\0.0183\end{bmatrix}$ |
| $\mathfrak{D}_1$ $\mathfrak{D}_2$ $\mathfrak{D}_3$ | $[0.5,0.55]$ | $[0.5,0.7]$ $[0.4,0.75]$ | $[0.5,0.7]$ $[0.4,0.85]$ | $[0.4,0.7]$ $[0.45,0.6]$ | $[0.4,0.55]$ $[0.45,0.5]$ $[0.4,0.53]$ |
| $\tau_1$ $\tau_2$ $\tau_3$ | 0.5 0.5 0.5 | 0.6 0.6 0.5 | 0.6 0.45 0.6 | 0.55 0.5 0.5 | 0.4 0.45 0.5 |
| State response | Figure 1 | Figure 2 | Figure 3 | Figure 4 | Figure 5 |

## 5. Conclusions

In this paper, we studied the stability and robust stability of SPLSs in which all subsystems are unstable by means of Φ-dependent dwell-time switching. By using the discretized co-positive Lyapunov functions, the sufficient conditions for the stability of SPLSs were obtained in the form of linear matrix inequalities. The stability and robust stability of SPLSs in continuous-time and discrete-time cases were studied in this paper, respectively. At the end of the paper, an illustrative example showed that the Φ-dependent dwell time technique is more effective than the mode-dependent dwell time and the dwell time.

**Author Contributions:** Conceptualization, Q.Y.; Methodology, Q.Y.; Software, X.J.; Validation, X.J.; Formal analysis, Q.Y.; Investigation, Q.Y.; Resources, X.J.; Data curation, X.J.; Writing—original draft, Q.Y. and X.J.; Writing—review & editing, Q.Y.; Visualization, X.J.; Supervision, Q.Y.; Project

administration, X.J.; Funding acquisition, Q.Y. All authors have read and agreed to the published version of the manuscript.

**Funding:** This work was supported by the Fundamental Research Program of Shanxi Province (202103021224249) and the Fund Program for the Scientific Activities of Selected Returned Overseas Professionals in Shanxi Province (20220023).

**Data Availability Statement:** The data supporting the reported results are available from the corresponding author upon reasonable request.

**Conflicts of Interest:** The authors declare that they do not have any commercial or associative interests that would represent a conflict of interests in connection with the work.

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
