# Peer review of "Robust Stability of Switched-Interval Positive Linear Systems with All Modes Unstable Using the Φ-Dependent Dwell Time Technique"

_axioms, doi:10.3390/axioms12070686_

Round 1

Reviewer 1 Report

The topic of the work is of great current interest in the scientific community. The authors study the stability of a positive linear system with switching using the approach of co-positive Lyapunov functions, Metzler matrices, and the Φ-dependent dwell time technique. The special character of the study is given by the consideration of the case in which all the subsystems are unstable. The difficulty consists in finding that switching law, with the related properties, which stabilizes the entire system.

1. The manuscript is, to some extent, a concentrated summary of other works by the authors (e.g. reference [23] or https://doi.org/10.1016/j.jfranklin.2020.07.052 - where the theoretical part is much more explained). The dense information written concisely makes the work quite difficult to be read, being necessary to search, at every step, for details about the notations used or the relationships written. A recommendation is to give consistency to the text, a natural fluency, with the explanation of the notations.

2. To what extent are the conditions of Theorem 1 fulfilled? What are the limitations?

3. The numerical example needs some more details, as well as the Figures and Table 2. For example, Table 2 provides far too much information to be covered in only 2 lines in the text.  Usually, the numerical example shows, in detailed steps, how the theoretical part is applied and it represents a consistent part of a paper.

4. The Figures are not cited in text.        

5. The authors are kindly asked to give some examples of real world system meeting those requirements (e.g. Metzler matrices, etc). In literature, most of the numerical examples are applied on simple cases (simple matrices). The real life applications deal with high order matrices where the stability conditions (especially the conservative ones) are hard, or even impossible, to be proved.

A proof read of the paper is recommended.

Reviewer 2 Report

The presented work is carried out at a high scientific level and contains new scientific results in the problems of analysis and synthesis of systems with switching.

The work is well organized, contains a meaningful and fairly complete overview of the problem. Theoretical results are substantiated by rigorous mathematical calculations. The simulation results confirm the obtained results and are compared with known approaches.

A fairly large amount of references on this topic is given.

Note on the numerical example.

Figures 1-3 are not meaningful. Looking at them, one can make a false conclusion that x1 variables are non-decreasing functions of time. In our opinion, one should either simply write out the eigenvalues of these systems or introduce additional initial conditions on x2 into the graphs in order to show decreasing portions of the x1 variable graphs. Apparently, such a situation should exist for at least one system.

Author Response

Dear Sir,

We would like to thank you for your evaluation of our paper. We have carefully revised the paper according to your comments and suggestions. The detailed responses are attached below.

Reviewer 2: The presented work is carried out at a high scientific level and contains new scientific results in the problems of analysis and synthesis of systems with switching. The work is well organized, contains a meaningful and fairly complete overview of the problem. Theoretical results are substantiated by rigorous mathematical calculations. The simulation results confirm the obtained results and are compared with known approaches. A fairly large amount of references on this topic is given.

 Note on the numerical example. Figures 1-3 are not meaningful. Looking at them, one can make a false conclusion that x1 variables are non-decreasing functions of time. In our opinion, one should either simply write out the eigenvalues of these systems or introduce additional initial conditions on x2 into the graphs in order to show decreasing portions of the x1 variable graphs. Apparently, such a situation should exist for at least one system.

Responses: Thank you very much for the reviewer's suggestions. We agree with your opinion. According to your comments, we have added those eigenvalues of subsystems to show the instability of subsystems and removed Figures 1-3 to eliminate their misleading effects. Your comment has great significance for this article.